# Pneumothorax, an Uncommon but Devastating Complication Following Shoulder Arthroscopy: Case Reports

**DOI:** 10.3390/medicina58111603

**Published:** 2022-11-05

**Authors:** Wei-Chien Sun, Liang-Tseng Kuo, Pei-An Yu, Cheng-Pang Yang, Huan Sheu, Hao-Che Tang, Yi-Sheng Chan, Alvin Chao-Yu Chen, Kuo-Yao Hsu, Chun-Jui Weng, You-Hung Cheng, Chih-Hao Chiu

**Affiliations:** 1Department of Orthopedic Surgery, Linkou Chang Gung Memorial Hospital, Taoyuan 333, Taiwan; 2Department of Orthopedic Surgery, Chiayi Chang Gung Memorial Hospital, Chiayi 613, Taiwan; 3Department of Orthopedic Surgery, Taoyuan Chang Gung Memorial Hospital, Taoyuan 333, Taiwan; 4Department of Orthopedic Surgery, Keelung Chang Gung Memorial Hospital, Keelung 204, Taiwan; 5Department of Orthopedic Surgery, New Taipei Municipal Tucheng Hospital, New Taipei City 236, Taiwan; 6Department of Orthopedic Surgery, Kaohsiung Municipal Feng-Shan Hospital, Kaohsiung 833, Taiwan

**Keywords:** shoulder arthroscopy, pneumothorax, interscalene nerve block, irrigation system, patient position

## Abstract

Shoulder arthroscopy is a mature and widely used treatment to deal with various shoulder disorders. It enables faster recovery and decreases postoperative complications. However, some complications related to shoulder arthroscopy cannot be neglected because they could be life threatening. We presented three cases of various clinical manifestations of pneumothorax after shoulder arthroscopy. The first case was a 65-year-old female who underwent arthroscopic rotator cuff repair under general anesthesia and interscalene nerve block in the beach-chair position. The second case was a 58-year-old male undergoing arthroscopic rotator cuff repair and reduction in glenoid fracture under general anesthesia in the lateral decubitus position. The third case was a 62-year-old man receiving arthroscopic rotator cuff repair under general anesthesia in the lateral decubitus position. Each case’s operation time was 90, 240, and 270 min. The pressure of the irrigation pumping system was 30, 50, and 70 mmHg, respectively. The second and third cases did not undergo interscalene nerve block. Although the incidence of pneumothorax following shoulder surgery and interscalene nerve block was only 0.2%, it is one of the most life-threatening complications following shoulder arthroscopy. In these cases, multifactorial factors, including patient positioning, interscalene nerve block, long surgical time, size of rotator cuff tears, and the pressure of the irrigation and suction system, can be attributed to the occurrence of pneumothorax. It is crucial to fully comprehend the diagnosis and management of pneumothorax to reduce the risk for patients receiving shoulder arthroscopy.

## 1. Introduction

Shoulder arthroscopy is widely used to treat various shoulder disorders, such as rotator cuff tears, shoulder instabilities, and fractures around the shoulder joint [1,2]. The safety of shoulder arthroscopy has been proven, while complications, including neurovascular injury (axillary, suprascapular, and musculocutaneous nerves), neurologic events (neurapraxia of the great auricular nerve or cranial nerve, injury to the cervical plexus), ophthalmoplegia, fatal venous air embolism, hypoperfusion stroke, infection, and thromboembolic events, were reported [3]. Different patient positioning during the surgery has exclusive complications [4]. Some of these complications might develop from insidious events to potentially severe clinical emergencies. We presented three cases of pneumothorax with different clinical presentations following shoulder arthroscopy. Among the three primary shoulder surgeons (Liang Tseng Kuo, Pei-An Yu, and Chih-Hao Chiu), they have operated on an average of 300, 150, and 400 shoulder arthroscopies annually. The objective of this study is to present their clinical manifestation of them and remind shoulder surgeons about this uncommon but life-threatening condition when it occurs. On the other hand, it provides evidence for further reference if another unwilling pneumothorax occurs.

## 2. Clinical Cases

### 2.1. Case 1

A 65-year-old female with hypertension, type II diabetes mellitus, and mitral valvular insufficiency presented with right supraspinatus and subscapularis tears in our clinic (Figure 1A). There was no known contributory traumatic event or smoking before, nor chest pathology (Figure 1B). She had an appendectomy, hemorrhoidectomy, and resection of bladder polyps with smooth recovery before. Her body mass index (BMI) was 26.7. After one year of failed medical treatment and rehabilitation, she decided to receive arthroscopic rotator cuff repair at our institute. Her American Society of Anesthesiologists (ASA) physical status classification system score was 3.

After general anesthesia under orotracheal intubation and interscalene block using an in-plane technique, the patient was put in the beach chair position for shoulder arthroscopy with a traction device (Spider, Smith & Nephew, MA, USA). The arterial oxygen saturation was around 95–96% before induction. Right supraspinatus and subscapularis repair with an independent-double-row technique [5], after an adequate rotator interval release (Figure 1C) and acromioplasty (Figure 1D) within 90 min with irrigation pressure of 30 mmHg. Given the bilateral symmetrical and clear breathing sound and clear consciousness without unstable hemodynamic status, the lung recruitment maneuver was performed, and she was extubated at SaO2 95%. The vital sign was stable during the whole surgical procedure. However, desaturation occurred 10 min after we finished the operation. In the postoperative recovery room, the patient gradually developed shortness of breath and presented chest tightness. The oxygen demand increased from a simple mask of 6 L per minute to a non-rebreathing mask of 15 L per minute within 2 h. The emergent chest anteroposterior radiograph revealed the right large pneumothorax with trachea and mediastinum left shift (Figure 1E). Consequently, the chest tube was placed smoothly without immediate complications (Figure 1F). The chest tube was removed on postoperative day 4, and she was discharged on postoperative day 5 without any other complications.

### 2.2. Case 2

A 58-year-old male without underlying diseases presented with right shoulder pain after falling down on the ground with a direct contusion of the same shoulder. The plain radiograph and computed tomography showed right anterior glenoid fossa fracture, Ideberg type Ia (Figure 2A,B). Ultrasonography revealed a full-thickness tear of the supraspinatus and subscapularis. His BMI was 26.7, and the ASA was 3. The preoperative chest X-ray was unremarkable (Figure 2C). Arthroscopic suture fixation for glenoid rim fracture with anchors, and supraspinatus and subscapularis repair by the single-row technique was performed in a lateral decubitus position under general anesthesia with orotracheal intubation and irrigation pressure of 70 mmHg. The operative time was around 4 h. The routine postoperative chest radiograph to check the position of suture anchors was implemented on postoperative day 1. However, right pneumothorax was found incidentally (Figure 2D). Although the vital sign was stable, without shortness of breath, a chest tube was placed by an experienced chest surgeon (Figure 2E). The follow-up chest radiography displayed improved right lung expansion, and the right chest tube was removed on postoperative day 5 without further complications (Figure 2F).

### 2.3. Case 3

A 62-year-old active male smoker had right shoulder pain with limited active range of motion for 6 months. He also had severe obstructive sleep apnea, hypertension, hyperlipidemia, and coronary artery disease status after percutaneous transluminal coronary angioplasty with one drug-eluting stent, colon diverticulitis, and right ureter stone. His BMI was 33.5. His ASA was 3. The preoperative chest X-ray was unremarkable (Figure 3A). On physical examination, the abduction and forward flexion was limited to 120 degrees, with a positive Hawkins test, belly press test, and Jobe’s test. Ultrasonography showed a right supraspinatus tear.

The patient was put in the lateral decubitus position under general anesthesia with orotracheal intubation, and arthroscopic rotator cuff repair was performed using suture anchors in a suture-bridge fashion [6] (Figure 3B) in 2 h with irrigation pressure of 70 mmHg. However, the decreased oxygen saturation from 99% to 90% with deteriorated systolic blood pressure from 139 to 89 and increased heart rate from 72 to 115 was found during the operation under pressure support ventilation. Decreased breathing sound was auscultated in the right chest. The portable chest radiography displayed large pneumothorax with mediastinum and trachea left shift (Figure 3C). The right tension pneumothorax was impressed, so the chest surgeon was consulted. Needle decompression was emergently performed. Right chest pigtail drainage was placed smoothly by the chest surgeon (Figure 3D). The oxygen saturation improved from 90% to 97% after the treatment. On postoperative day 5, the right chest pigtail was removed due to improved right lung expansion and symmetrical chest wall expansion without shortness of breath. The patient was discharged smoothly.

## 3. Discussion

Although shoulder arthroscopy is a safe and effective surgical procedure, the rare postoperative complications cannot be neglected. Some of the complications can be potentially life threatening, such as pneumothorax. In our cases, there are some potential causes of perioperative pneumothorax, as follows:1Patient positioning

Pneumothorax can occur in patients with beach-chair position and lateral decubitus position. There are three studies in which the patients presented with pneumothorax after surgery in the lateral decubitus position under general anesthesia without an interscalene nerve block [7,8,9]. One of these studies assumed that continuous axial traction can increase the laxity of the junction around the shoulder and chest wall, which might cause pneumothorax [9]. The traction force can increase the space in the glenohumeral joint and subacromial space and cause neurovascular and soft-tissue injury [4]. The lateral decubitus position makes the operative side higher and causes more negative pleural pressure than the nonoperative side. The pressure gradient between the positive pressure of anesthesia and lateral decubitus position may put the patient at risk of a higher pressure gradient, which could cause alveolar rupture [10]. However, one of the pneumothoraxes in our group had surgery in the beach-chair position with traction. Therefore, patients in both surgical positions should be watched carefully.

2Interscalene nerve block

The interscalene nerve block provides effective anesthesia for most shoulder surgeries, including arthroscopic procedures, arthroplasty, and fracture fixation, and showed excellent successful rates of about 97% and a relatively low complication rate of about 2.3% without major complications, such as seizure, pneumothorax, cardiac event, or thromboembolism [11]. However, the out-of-plane technique of interscalene nerve block may result in poor needle tip visualization, which could disrupt or puncture the parietal or visceral pleura. Hence, the success of ultrasound-guided interscalene nerve block is highly operator dependent [12,13]. Moreover, the patient factor, including lung diseases, smoking, emphysema, and obesity, may increase the difficulty of implementing local and general anesthesia [13,14].

3Surgical time

Longer surgical time exposed patients to a burden of more risk of adverse effects from patient positioning, arthroscopic procedures, and anesthesia. However, there was no evidence yet.

4The size of rotator cuff tear

There is still debate as to whether the extensive release during shoulder arthroscopy causes pneumothorax. There was one study demonstrating that a 61-year-old female with a massive rotator cuff tear who had no history of smoking, COPD, asthma, or other pulmonary diseases underwent arthroscopic repair and aggressive release of rotator interval by intra-articular shaving under general anesthesia without local regional nerve block [15]. In our clinical practice, extensive release and subacromial decompression were performed elegantly and skillfully. The dangerous area was nearby the coracoid base when performing subacromial decompression.

5Irrigation system

One study indicated that the change of subacromial pressure gradient attributed to intermittent use of suction and continuous infusion pump may cause pneumothorax following shoulder arthroscopy with the use of an irrigation system [7]. The air could enter the shoulder joint via other portals if the fluid pressure of the glenohumeral and subacromial joint was not enough to expand the joint space. The air may be entrapped or accumulated in the shoulder joint or extending other spaces via fascia or axillary sheath [16].

### Management

In our case series, the timing of diagnosis of pneumothorax was different in each case. As for the first case, suspicion of pneumothorax was raised at the end of the operation due to relatively decreased oxygen saturation, and the emergent chest tube placement was performed 2 h after the operation. As for the second case, the patient did not present obvious shortness of breath and chest tightness, and the right large pneumothorax was found by routine chest radiography for checking the position of the suture anchors on postoperative day 1. As for the third case, the tension pneumothorax was found intraoperatively, and emergent needle decompression and chest pigtail catheter were placed immediately.

The management of pneumothorax depends on the size, the severity of symptoms, whether there is a persistent air leak, and whether the pneumothorax is primary or secondary [17]. According to the British Thoracic Society, pleural disease guideline 2010 for treatment of spontaneous pneumothorax, tension pneumothorax, bilateral pneumothorax, or unstable hemodynamic status were suggested for chest tube insertion. Reviewing the patient’s age, evidence of underlying lung diseases, chest radiograph, and symptoms can determine the corresponding treatment, including observation, needle aspiration, small-bore chest drain, large-bore chest tube, oxygen supplement, and whether admission is necessary [18]. The observation and follow-up can be conducted in the outpatient clinic for small primary pneumothorax without symptoms. The needle (16–18 G) aspiration is indicated for primary pneumothorax with a size > 2 cm and/or breathless. The small-bore (8–14 Fr) chest drain and admission are indicated for the secondary pneumothorax with a size > 2 cm or breathless.

## 4. Conclusions

Although the incidence is rare (about 0.2%), pneumothorax is one of the most life-threatening complications following shoulder arthroscopy. Multiple factors, including patient positioning, interscalene nerve block, surgical time, the size of rotator cuff tear, and the use of an irrigation system, can be attributed to the cause of pneumothorax. It is crucial for shoulder surgeons to raise suspensions and fully comprehend the diagnosis and management of pneumothorax.

## Figures and Tables

**Figure 1 medicina-58-01603-f001:**
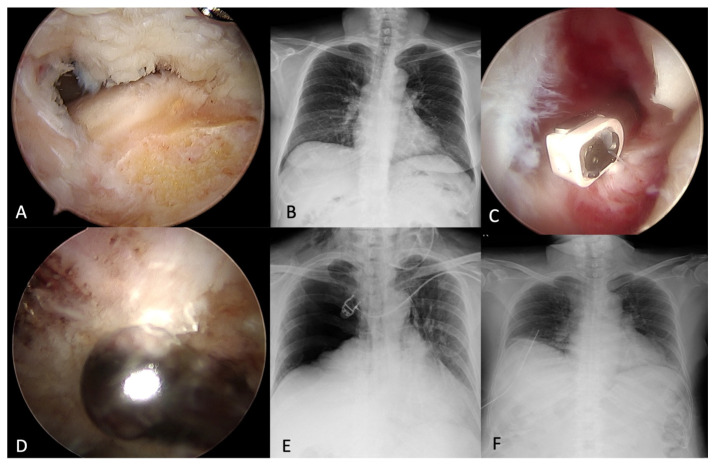
A 65-year-old female with (**A**) Right supraspinatus and subscapularis tears. (**B**) Preoperative chest X-ray was unremarkable. (**C**) Rotator interval release. (**D**) Acromioplasty. (**E**) Right pneumothorax with trachea and mediastinum left shift. (**F**) A chest tube was placed smoothly without complications.

**Figure 2 medicina-58-01603-f002:**
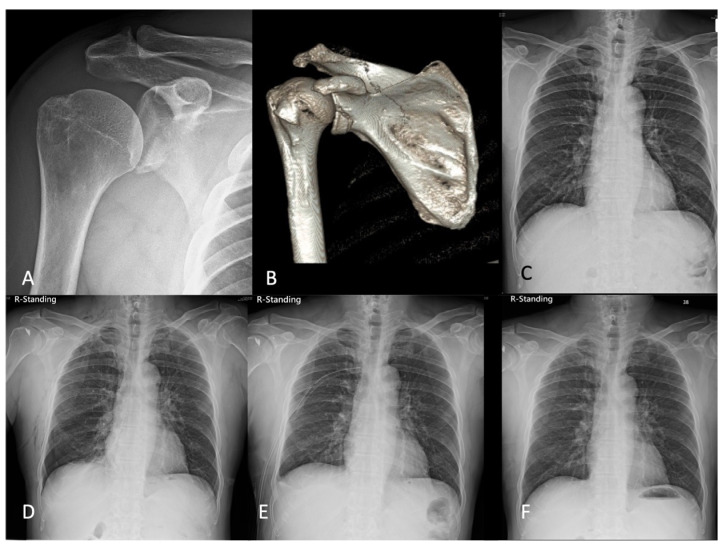
A 58-year-old male with (**A**,**B**) Right anterior glenoid fossa fracture, Ideberg type Ia. (**C**) Preoperative chest X-ray was unremarkable. (**D**) Right pneumothorax was found incidentally. (**E**) A chest tube was placed. (**F**) Follow-up chest radiography displayed improved right lung expansion.

**Figure 3 medicina-58-01603-f003:**
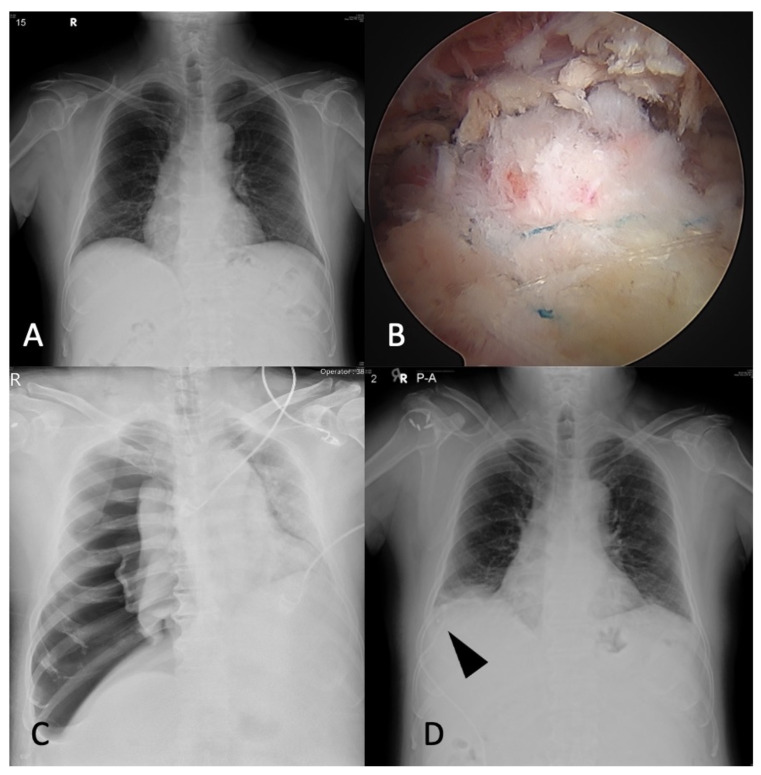
A 62-year-old male patient with (**A**) Preoperative chest X-ray was unremarkable. (**B**) Suture-bridge technique for right rotator cuff tear. (**C**) A large pneumothorax with mediastinum and trachea left shift after the surgery. (**D**) A chest pigtail drainage was placed smoothly. Arrowhead: pigtail drainage.

## Data Availability

Not applicable.

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
