# Peer review of "Pneumothorax, an Uncommon but Devastating Complication Following Shoulder Arthroscopy: Case Reports"

_medicina, 2022, doi:10.3390/medicina58111603_

Round 1

Reviewer 1 Report

The work deals with a rare but potentially severe complication of shoulder arthroscopy, pneumothorax. The paper does not present improprieties but is not original and does not fit into a wider series (for example the three cases in how long they occurred and on how many arthroscopies in a year? Were the operators all experts?). 

I also do not understand if these three patients had predisposing factors to the development of a pneumothorax and if some precautions would have avoided this complication.

I believe that the bibliography cited on the subject is also limited.

Author Response

The work deals with a rare but potentially severe complication of shoulder arthroscopy, pneumothorax. The paper does not present improprieties but is not original and does not fit into a wider series (for example the three cases in how long they occurred and on how many arthroscopies in a year? Were the operators all experts?). 

Reply: Thanks for the comment. Since pneumothorax is a rare complication following shoulder arthroscopy, it’s challenging to collect enough cases to fulfill a wider series. Among the three primary shoulder surgeons (Liang Tseng Kuo, Pei-An Yu, and Chih-Hao Chiu), they have operated on an average 300, 150, and 400 shoulder arthroscopies annually. The above detail is added in the revised manuscript.

I also do not understand if these three patients had predisposing factors to the development of a pneumothorax and if some precautions would have avoided this complication.

Reply: Thanks for the comment. Since pneumothorax is a rare complication following shoulder arthroscopy, we have a paucity of evidence, for now, to tell the precautions to avoid this complication because of the low case numbers. The objective of this study is to present their clinical manifestation of them and remind shoulder surgeons about this uncommon but life-threatening condition once happened. On the other hand, it provides evidence for further reference if another unwilling pneumothorax once happened. The above detail is added in the revised manuscript.

Reviewer 2 Report

The authors describe an uncommon complication of shoulder surgery. Currently, there are few reported cases in the literature.

The manuscript is interesting and novel but can be improved.

Comments:

  • The biggest problem of the article is that the causes related to the patient are not mentioned. We know that the Procedure-, patient-, and operator-related factors determine the likelihood of iatrogenic pneumothorax. In the paper, the authors only mentioned the aspects related to shoulder surgery. But, Patient-related factors should also be mentioned ( For example, Underlying lung disease: severe emphysema, bullous lung disease, or patients body mass index). Also, barotrauma or Barotrauma in mechanically ventilated patients should be considered. 
  • In line 53: valvular insufficiency. In Which valve? 
  • In line 55: Change "There was no known contributory traumatic event or smoking before and chest pathology" . Correct. 
  • In line 152 there is a writing error. Correct. 
  • It would be interesting to know if the procedure was performed under orotracheal intubation or laryngeal mask

Author Response

The authors describe an uncommon complication of shoulder surgery. Currently, there are few reported cases in the literature. The manuscript is interesting and novel but can be improved.

Comments:

  • The biggest problem of the article is that the causes related to the patient are not mentioned. We know that the Procedure-, patient-, and operator-related factors determine the likelihood of iatrogenic pneumothorax. In the paper, the authors only mentioned the aspects related to shoulder surgery. But, Patient-related factors should also be mentioned (For example, Underlying lung disease: severe emphysema, bullous lung disease, or patients body mass index). Also, barotrauma or Barotrauma in mechanically ventilated patients should be considered. 

Reply: Thanks for the comment. The BMI for each case was 26.7, 26.7, and 33.5, which is mentioned in the manuscript. As most of the primary spontaneous pneumothorax happened in patients with low BMI (<18 kg/m2) in the Chinese population[1], the three patients in our series had two overweight (BMI >25 kg/m2) and one obese (BMI >30 kg/m2) condition, which is contradictory to the published evidence. However, due to the low case number of the series, it is difficult to conclude that higher BMI is related to the risk of pneumothorax following shoulder arthroscopies.

  • In line 53: valvular insufficiency. In Which valve? 

Reply: Thanks for the comment. It was a mitral valvular insufficiency. The above detail is added in the revised manuscript.

  • In line 55: Change "There was no known contributory traumatic event or smoking before and chest pathology" . Correct. 

Reply: Thanks for the comment. The sentence is corrected into “There was no known contributory traumatic event or smoking before, nor chest pathology”.

  • In line 152 there is a writing error. Correct. 

Reply: Thanks for the comment. The sentence is modified into “The pressure gradient between the positive pressure of anesthesia and lateral decubitus position may be prone to the patient to risk a higher-pressure gradient that could cause the rupture of alveolar rupture.[10] However, one of the pneumothoraxes in our group had surgery in the beach-chair position with traction” in the revised manuscript.  

  • It would be interesting to know if the procedure was performed under orotracheal intubation or laryngeal mask

Reply: Thanks for the comment. All surgeries were performed under orotracheal intubation. The above detail is added in the revised manuscript.